# Numerical Study on the Waterjet–Hull Interaction of a Free-Running Catamaran

**Yanlin Zou [1], Dakui Feng [1,*], Weihua Deng [1], Jun Yang [1] and Hang Zhang [2]**

[1] Key Laboratory of Ship and Ocean Hydrodynamics of Hubei Province, School of Naval Architecture & Ocean Engineering, Huazhong University of Science and Technology, Wuhan 430074, China; zouyanlin@hust.edu.cn (Y.Z.); zhua_@hust.edu.cn (W.D.); yangjun98@hust.edu.cn (J.Y.)

[2] Hubei Institute of Aerospace Chemical Technology, Xiangyang 441003, China; zhanghang@casc42.cn

\* Correspondence: feng_dk@hust.edu.cn

**Abstract:** Waterjet–hull interaction is the hot point and research focus in the research of waterjet-propelled crafts. This paper presents numerical studies on the interaction between a waterjet system and a catamaran. Numerical simulations of both bare hull and self-propulsion hull were carried out based on the URANS method. The SST k-ω model is selected for the closure of the URANS equations. The level set method together with the dynamic overset grid approach is used for the simulations. The body force model with the PI speed controller is used to simulate the rotational motion of the rotor in the simulations for the self-propulsion hull. Moreover, uncertainty analyses of the numerical method are conducted to verify the accuracy of the numerical solver. The numerical results of the bare hull and self-propulsion hull are compared in detail, such as the wave pattern, pressure distribution, hull attitude, and so on. The waterjet reduces the pressure on the hull surface near the stern and makes the height of the wave near the stern lower. This leads to a more violent change in hull attitude and the thrust deduction is positive, ranging from 0.1 to 0.2. The energy conversion is analyzed based on the ITTC recommended procedures, which shows the overall efficiency of the waterjet behind the hull is about 0.75~0.8 times the free stream efficiency.

**Keywords:** URANS; waterjet; catamaran; waterjet–hull interaction; body force model





## 1. Introduction

The waterjet propulsion system is a widely used device on high-speed vehicles. Compared with a conventional propeller, the waterjet has higher efficiency and better anti-cavitation characteristics. In addition, this propulsion mode can be adapted to shallow water conditions.

With the development of waterjet technology, evaluation of the waterjet propulsion systems has attracted attention gradually. Fujisawa [1] described experimental techniques for the waterjet propulsion system in a water tunnel. The measured performances of the pump and propulsion of the model system have great agreement with the field experiment with prototype craft. It suggests that the nozzle diameter should be decreased to obtain higher efficiency in a wide range of craft velocities. Seo et al. [2,3] conducted a model test to predict the propulsion performance of amphibious ship in water. The energy conversion efficiency of water jet propulsion device has been evaluated and the interaction between hull and waterjet was also studied in detail. Gong et al. [4] used Particle Image Velocimetry (PIV) technology to measure the inlet velocity distribution of the waterjet propulsion ship. The influence of the ship type and the shape of the flow area on the efficiency of the waterjet propulsion was analyzed in detail. Moreover, there is a large deviation between the theoretical efficiency and the measured efficiency in the waterjet design and application. Huang et al. [5] also used PIV technology to study the flow characteristics in the inlet duct of the water jet propulsion device. It was found that the hydraulic loss in the inlet duct increased gradually with the increase in flow rate. It was also found that there would be a

recirculation region near the duct lower wall with a high-velocity flow near the upper wall. Then, a shear flow with an obvious velocity gradient presented in the horizontal straight part of the pipe.

With the development of Computational Fluid Dynamics (CFDs) technology, the reliability of numerical simulation method becomes higher and higher, and it is widely used in three-dimensional flow research. Compared with the model test, numerical simulation can obtain more flow field information, which is a very suitable method for waterjet propulsion system research. Song et al. [6] used a combined EFD-CFD approach to analyze the effects of interceptor installation on the velocity distribution around a waterjet. The PIV measurement results have great agreement with Reynolds-Averaged Navier–Stokes (RANS) simulations in general. Both of them can capture the velocity characteristics around the waterjet. Cao et al. [7] adopted CFD method to analyze the reason for the poor efficiency and found that the non-uniform inflow is the main reason for the drop of the propulsion efficiency.

When a waterjet propulsion system is applied on the hull, the waterjet–hull interaction is an important topic in the research of waterjet-propelled ships. Gong et al. [8] used a combined experimental and numerical approach to study a four-waterjet-propelled ship. It described how the propulsion performance is influenced by the waterjet–hull interaction for a self-propelled ship. Guo et al. [9] performed a series of simulations on a self-propulsion trimaran with a waterjet propeller. A Multiple Reference Frames (MRFs) model was used to replace the pump effect. The capture area was obtained by a numerical tool, which was very practical and could simplify the post-processing greatly. Jiang and Ding [10] analyzed the waterjet–hull interaction of a high-speed planning trimaran by using numerical simulations and experimental research. The thrust deduction and interaction efficiency were also investigated. Moreover, Eslamdoost [11,12] also investigated the waterjet–hull interaction in detail.

A waterjet system is a complex device compared with a conventional propeller. Many achievements in numerical prediction of hydrodynamic performance of waterjet propulsion were achieved. For the simulation of a waterjet, a short time step but a great number of grids are needed to describe the behavior of the rotor. As a result, some scholars use simplified methods to describe the behavior of the rotor. Eslamdoost et al. [13] presented a robust and fast method to study the waterjet–hull interaction. This method, also called the pressure jump method, was conducted by adding a constant pressure on the impeller disk. The method was validated by comparing the predicted results and measured data. Gong et al. [14] performed simulations of a ship model using the virtual disk and real impeller. The results revealed that the hull flow field coincided with each other for the two methods but there was an obvious difference in the internal flow field. There was significant consistency both in the local pressure distribution and wave jamming effect caused by jet stream. Eslamdoost and Vikström [15] conducted simulations to model a waterjet pump in axisymmetric inflow with different sophistication levels models. By comparing the results of different body force models, it can be known that the flow structures at the nozzle are mainly affected by the stator and less affected by the rotor. Therefore, when concentrating on the waterjet–hull interaction effects rather than the flow inside the pump, it is feasible to use body force model to represent the effects of the pump. Zhang et al. [16] applied virtual disks to a self-propulsion model and researched the stern flap-waterjet–hull interaction. Different stern flaps were investigated, and the mechanism of stern flap–waterjet–hull interaction was preliminarily proposed. According to their research, it is reasonable to replace the rotor with a virtual disk when the flow inside the duct is not mainly concerned.

To summarize, when designing and taking performance evaluation of waterjet ships, it is very significant to conduct research on the waterjet–hull interaction. Currently, varieties of new ship types are applied to high-performance ships, among which the catamaran has attracted attention because of its excellent maneuverability, transverse stability, good seakeeping performance, and so on. In this study, series of numerical studies of the interaction mechanism between a waterjet system and a catamaran are conducted by CFD

technology. The simulations of both bare hull and waterjet-propelled hull are conducted by using an in-house code, and uncertainty analyses of the numerical method for the bare hull are conducted to verify the accuracy of the numerical solver. The numerical results of the bare hull and self-propulsion hull are compared in detail. Moreover, the energy conversion of the waterjet behind the hull are calculated according to the International Towing Tank Conference (ITTC) guidelines. The main content of this paper is shown in Figure 1.

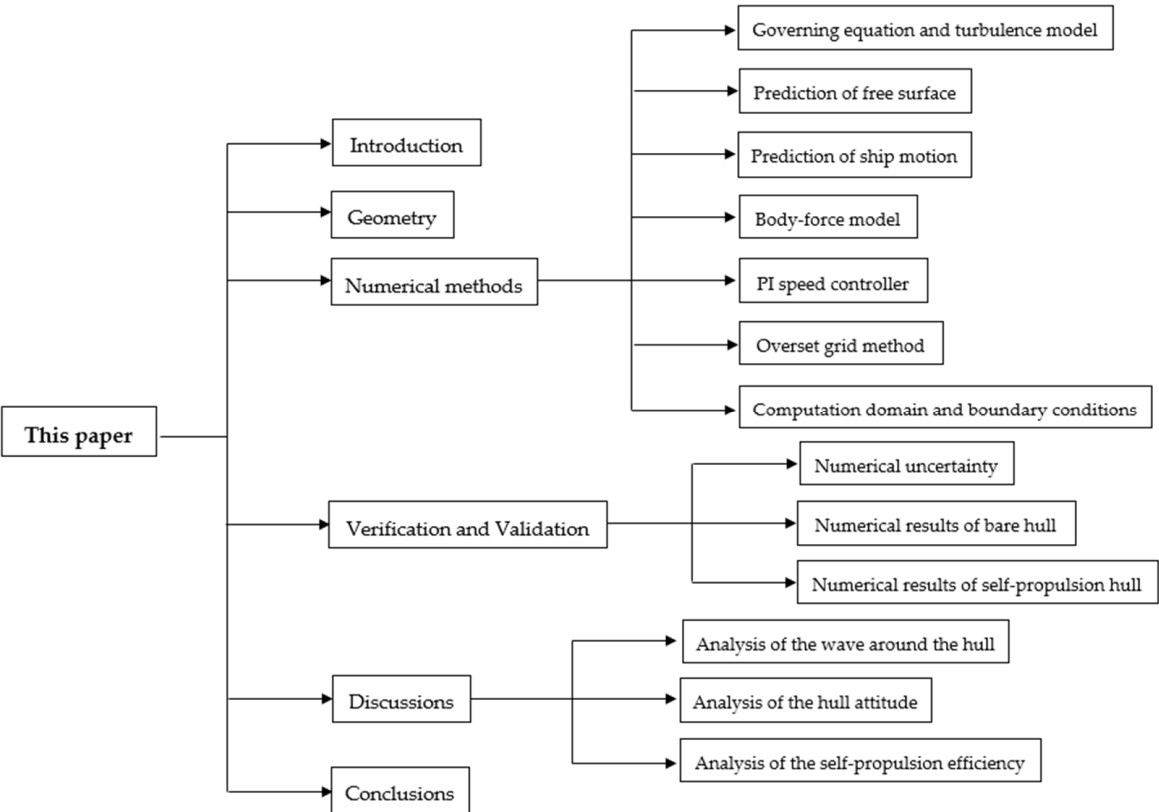

**Figure 1.** The main content of this paper.

## 2. Geometry

### 2.1. Hull Geometry

The Delft catamaran is investigated in this research. The waterjet propulsion system is used as the propulsion power of the hull. The Froude number at the design speed is 0.5. The main parameters of the model-scale Delft catamaran are presented in Table 1. The hull geometry is shown in Figure 2.

**Table 1.** The main parameter of the Delft catamaran.

| Main Parameter | Symbol | Value |
|---|---|---|
| Length between perpendiculars | $L_{PP}/m$ | 3.627 |
| Waterline length | $L_{WL}/m$ | 3.627 |
| Molded breadth | $B/m$ | 1.157 |
| Breadth of demihull | $b/m$ | 0.2904 |
| Demihull spacing | $s/m$ | 0.8470 |
| Bow draft | $T_F/m$ | 0.1815 |
| Stern draft | $T_A/m$ | 0.1815 |
| Volume of displacement | $\Delta/m^3$ | 0.07700 |
| Longitudinal center of gravity | $L_{CG}/m$ | 1.911 |
| Vertical center of gravity | $K_G/L_{PP}$ | 0.02715 |

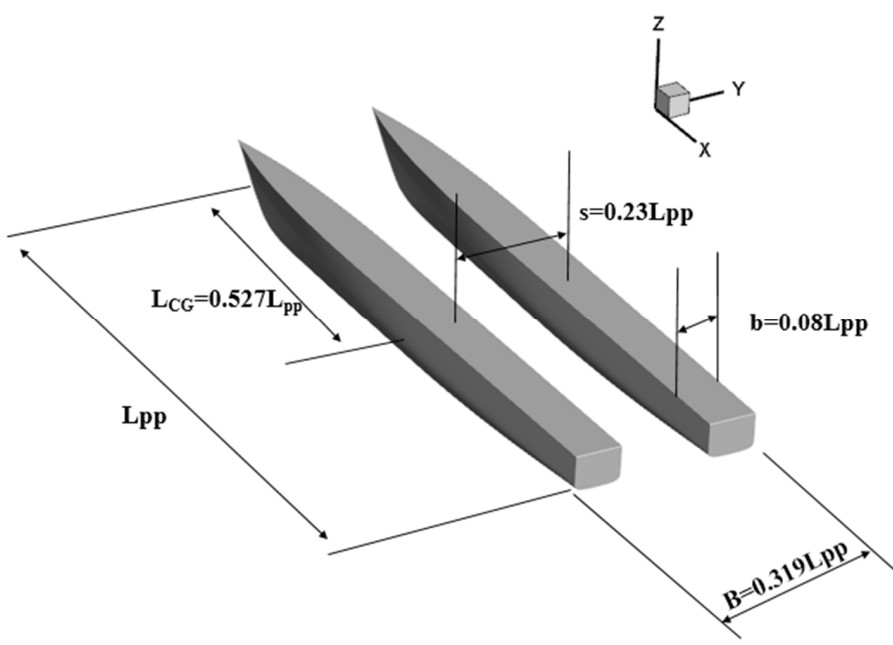

**Figure 2.** The geometry of the Delft catamaran.

*2.2. Waterjet Geometry*

Figure 3 provides the geometry of the demihull and the waterjet system. Compared with the bare hull, a waterjet system, including duct, stator, shaft, and rotor is added here. The parameter of the waterjet is shown in Table 2.

**Table 2.** Geometry parameter of the waterjet.

| Parameter of the Waterjet | Value |
|---|---|
| Diameter of the rotor ($D_R$)/m | 0.120 |
| Blades number of the rotor | 3 |
| Blades number of the stator | 8 |
| Tip clearance between rotor and duct/mm | 0.917 |
| Diameter of the nozzle/m | 0.0610 |
| Length of the duct/m | 0.800 |

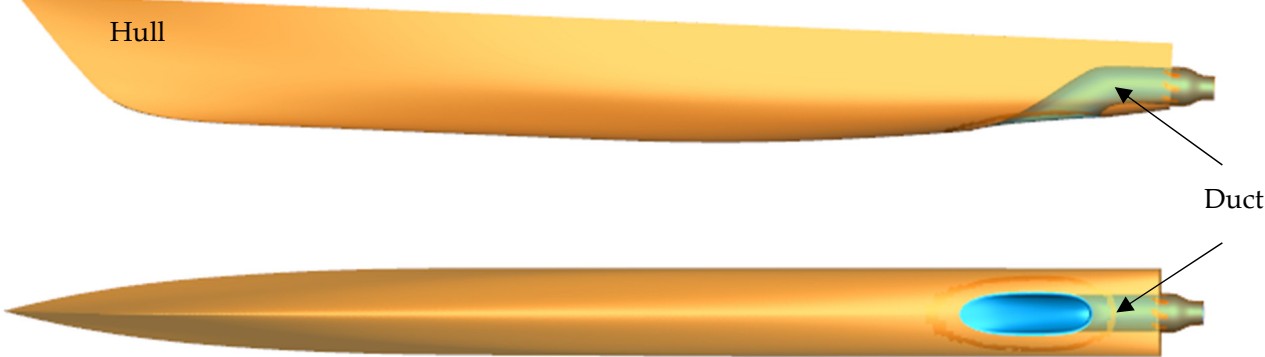

**Figure 3.** The demihull for the self-propulsion simulations.

### 3. Numerical Methods

*3.1. Governing Equation and Turbulence Model*

An in-house code is used for the simulations in this study. The accuracy of the in-house code has been verified and it was applied in the simulation of the self-propelled hulls [17–19]. It solves the unsteady and incompressible RANS equations stated as Equations (1) and (2):

$$\frac{\partial U_i}{\partial x_i} = 0 \tag{1}$$

$$\frac{\partial U_i}{\partial t} + U_j \frac{\partial U_i}{\partial x_j} = -\frac{1}{\rho} \frac{\partial \overline{p}}{\partial x_i} + \frac{1}{\rho} \frac{\partial}{\partial x_j} \left( \mu \frac{\partial U_i}{\partial x_j} - \rho \overline{u_i u_j} \right) + f_{bi} \tag{2}$$

In the above two equations, $U_i = (U_1, U_2, U_3)$ is the components of the Reynolds average speed. $\overline{p}$ denotes the time-averaged pressure. $\rho$ is the fluid density. $x_i = (x_1, x_2, x_3)$ represents the directions of the coordinate and $\mu$ is the dynamic viscosity. $\overline{u_i' u_j'}$ indicates Reynolds stress tensor. $f_{bi} = \left( f_{bx}, f_{by}, f_{bz} \right)$ is the term for the body force model which is taken as zero when the body force model is not used.

The Shear Stress Transport (SST) k-ω turbulence model put forward by Menter [20] is selected for the closure of the RANS equations. The SST k-ω turbulence model combines the advantages of standard k-ω turbulence model and standard k-ε turbulence model. For the solution of the near-wall region, it uses the standard k-ω turbulence model, while uses the standard k-ε turbulence model for the external boundary. Moreover, this model is applicable to solve the problem of boundary layer with adverse pressure gradient, and it has high accuracy in predicting flow separation and complex flow field. The finite difference method is applied to discretize the governing equations, and the PISO algorithm is used to solve the velocity–pressure coupling. The dynamic overset approach is employed to generate the grid of the catamaran and the waterjet.

*3.2. Prediction of the Free Surface*

In the simulations, the level set method [21] is applied to distinguish the free surface. A distance function $\phi$ is defined and it satisfies the following equation:

$$\frac{\partial \phi}{\partial t} + v \cdot \nabla \phi = 0 \tag{3}$$

In the equation, $v$ represents the velocity vector at any point in the flow field. The underwater point is defined when $\phi$ is positive, while the point in the air is defined when $\phi$ is negative. Therefore, the isosurface for $\phi = 0$ is the free water surface.

*3.3. Prediction of the Ship Motion*

In the simulations, the catamaran is regarded as a rigid object, and it has six degrees of freedom. There are two different coordinate systems defined in the solver, that is the Earth-fixed coordinate system and the ship-fixed coordinate system, respectively. The ship-fixed coordinate system is fixed at the gravity center of the catamaran, and it moves with the hull. The RANS equations are solved in the Earth-fixed coordinate system while the six degrees of freedom dynamic equations are solved in the ship-fixed coordinate system. The detailed equations of motion refer to Zhang et al. [22].

In this study, the motion of the catamaran is limited to three degrees of freedom translation along X, Z directions and rotation around Y direction in Figure 1.

*3.4. Body Force Model*

When simulating a self-propulsion catamaran, it costs a lot of computing resources to perform the rotation of the rotor because of the limitation of the time step. According to Eslamdoost and Vikström [15], when concentrating on the waterjet–hull interaction effects

rather than the inside flow of the pump, it is feasible to simulate the pump effect of the rotor by using body force model. Thus, to save computing resources and computing time, the rotor is replaced by a body force model which is suited at the location of rotor. The region of the body force model is a cylindrical area based on the chord length and diameter of the rotor. The sketch of the simplified waterjet system is shown in Figure 4.

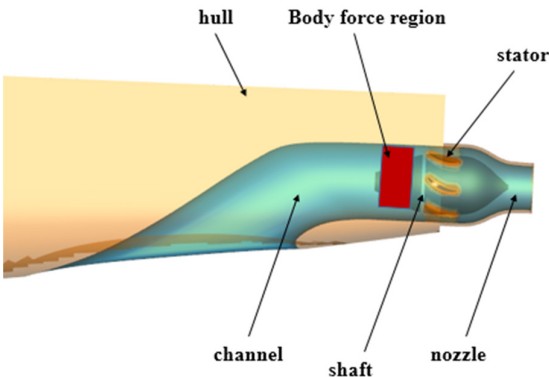

**Figure 4.** The simplified waterjet system and the location of body force model.

For the body force model, the performance of the rotor is applied to the disk as polynomial fit curves. The polynomial fit curves represent the relationship between thrust coefficient (or torque coefficient) and advance ratio. The advance ratio is calculated by the average velocity on the disk and the revolution rate of the disk. To obtain the polynomial fit curves, simulations for the open-water pump are conducted. The polynomial curves in Figure 5 are applied in the body force model.

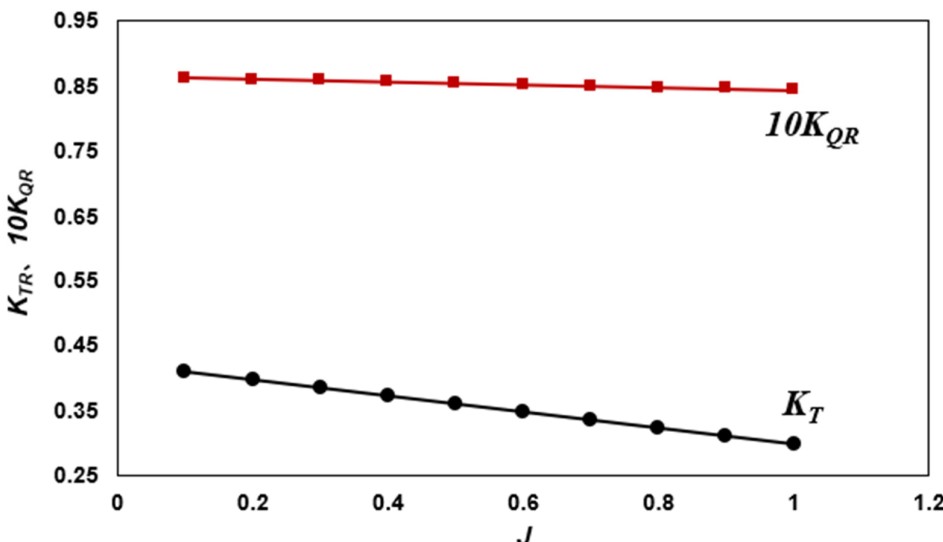

**Figure 5.** The open-water characteristic of the pump.

*3.5. PI speed Controller*

The revolution rate of rotor and hull speed are matched by proportional-integral (PI) speed controller [23]. In the self-propulsion simulations, the real-time revolution rate of rotor is calculated by the equation:

$$n = Pe + I \int_0^t e\, dt \qquad (4)$$

*P* is the proportional factor, and *I* is the integral factor. The difference between the target hull speed and current hull speed can be defined as:

$$e = U_{target} - U \tag{5}$$

With the iteration of the numerical simulations, the value of *n*, *U* and *e* changes gradually according to the target hull speed. Finally, the target velocity is achieved, and the revolution rate of the rotor is obtained.

### 3.6. Overset Grid Method

The structured grid is used for discretizing the computational domain. The catamaran with the waterjet system is too complex to generate the grids as a whole. Thus, the overset grid is employed. The overset grid method makes it possible to generate grids as hull, duct, nozzle, shaft, and stator. The relationship among these grids is established through the in-house overset program and the boundary information of grids is communicated with each other.

The overset grid method includes three steps: hole cutting, fringe nodes identification, and identification of the donor cells, respectively. The hole mapping method [24] is used for the hole cutting process to remove the unnecessary nodes inside of a solid surface. The cut–paste algorithm [25] is employed to generate the overset area. The donors of interpolation points are searched by alternating digital tree (ADT) method and the flow information of the fringe nodes is obtained by trilinear interpolation method [26]. Based on the different size between donor nodes and fringe nodes, an optimization is conducted to make at least two layers of grids participating in overset operation.

The dynamic overset grid method can deal with the relative movement between adjacent grids. It is suitable for the rotation simulation of rotor. Figure 6 shows the effect of the overset between stator and shaft as well as the overset between demihull and waterjet. The grid for the bare hull consists of background and demihull. The fully structured overset grid method is used, and the shape of the grid is hexahedral mesh. For the self-propulsion catamaran, the grid includes background grid, hull grid, and the propeller grid (nozzle grid, duct grid, shaft grid, and stator grid, etc.). The grid of the background and the hull are the same as them in the simulations of bare hull. The overset grid system of the self-propulsion demihull is shown as Figure 6c.

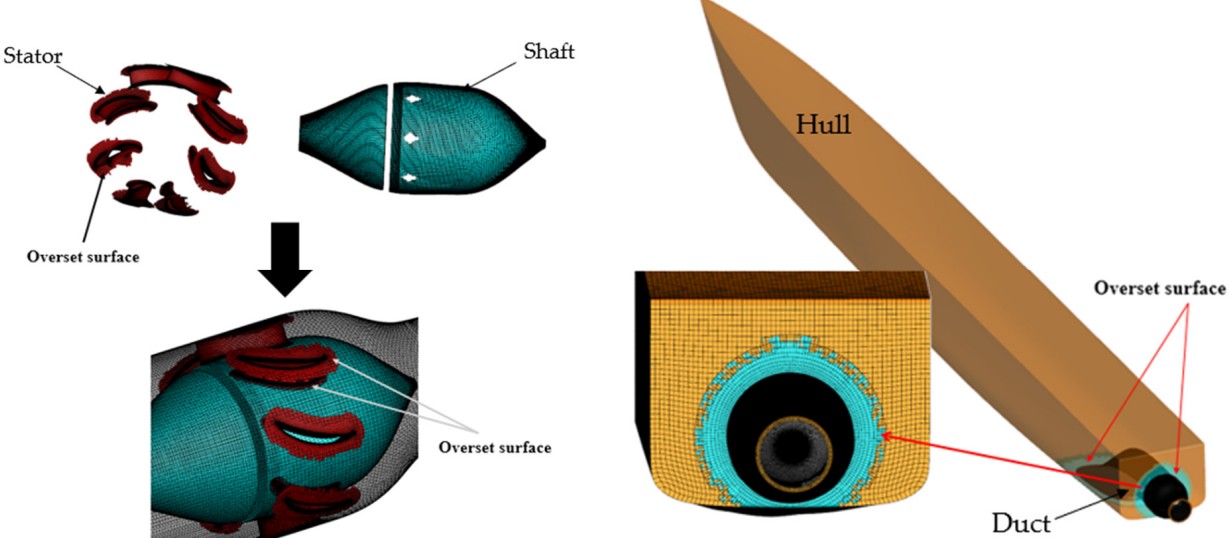

(**a**) The sketch of the overset process.

**Figure 6.** *Cont.*

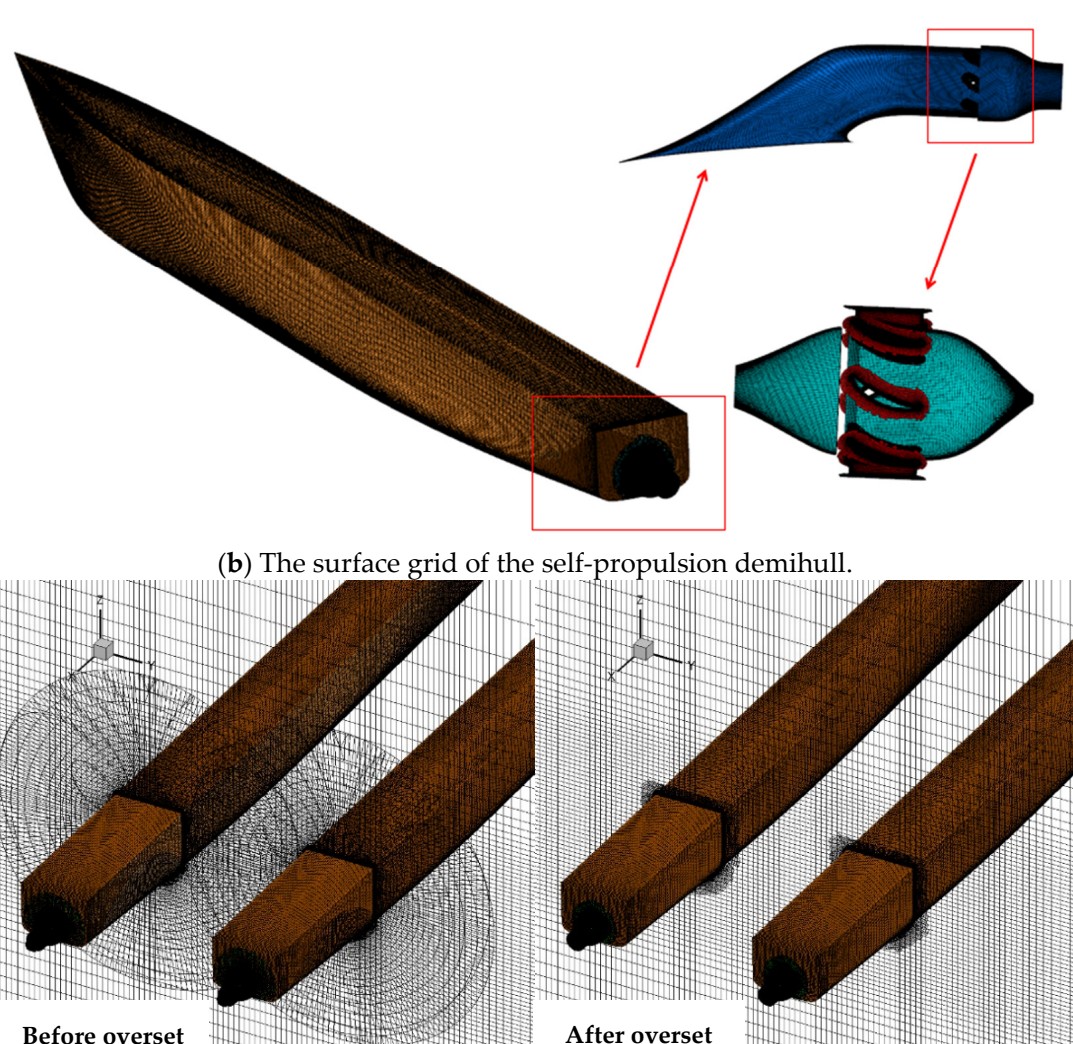

(**b**) The surface grid of the self-propulsion demihull.

Before overset

After overset

(**c**) Overset grid

**Figure 6.** Overset grid of the self-propulsion demihull.

### 3.7. Computational Domain and Boundary Conditions

Figure 7 shows the computational domain and boundary condition. It is a cuboid surrounding the catamaran. The two end surfaces are defined as velocity inlet and pressure outlet, respectively. According to the ITTC recommendation [27], the inlet boundary should be placed 1–2 $L_{pp}$ upstream, while the outlet boundary must be placed 3–5 $L_{pp}$ downstream.

The initial velocity of the flow is set at the velocity inlet. The surface of the hull is set as the no-slip wall. The longitudinal section in the center plane of the catamaran is defined as symmetry. In numerical modeling, half of the catamaran on one side of the symmetry plane is considered to save the computing resources on the premise of ensuring the accuracy.

In this paper, parallel computations are used to carry out numerical simulations with grid decomposition. The present simulations were performed based on the Beijing Supercomputer in Jinan with a parallel computational technique. A total of 56 cores are included for each processor (Intel® Xeon® Gold 6258R Processor@2.7 GHz). For the simulation of the bare hull, each simulation used a 56-core CPU, and took a physical time of about 12 h to convergence. Numerical simulation for the bare hull consumed a total of about 8448 cores. For the simulation of self-propulsion hull, each simulation used a 112-core CPU, and took a physical time of about 36 h to convergence. Numerical simulation for the self-propulsion hull consumed a total of about 32,256 cores.

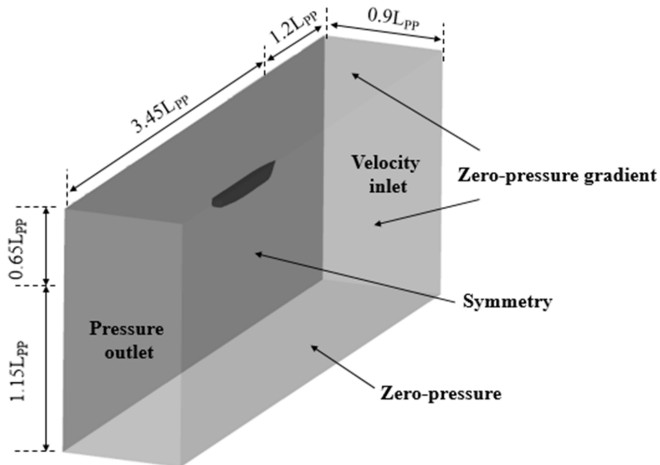

**Figure 7.** Computational domain and boundary condition.

## 4. Verification and Validation

### 4.1. Numerical Uncertainty

To ensure the accuracy of the simulations, the uncertainty analysis of the grid and time step are carried out at Fn = 0.5 according to the procedures [28,29]. The resistance coefficient, sinkage (at the center of gravity) and trim are taken as indexes of uncertainty. The experimental data are from the INSEAN. Sensitivity analysis three sets of grids and time steps are provided in Figure 8.

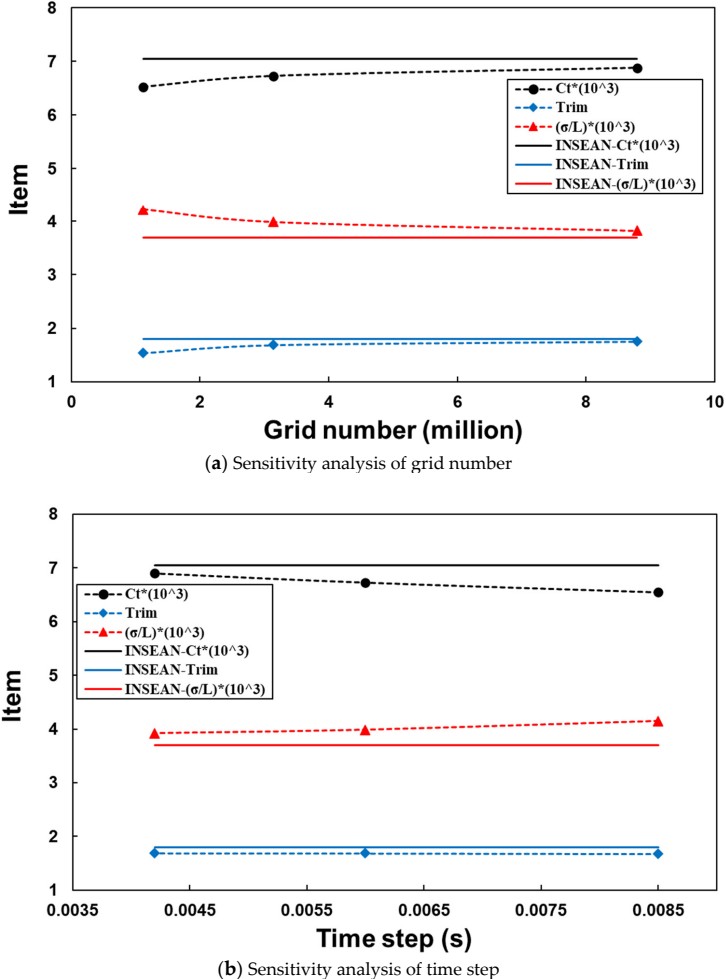

(**a**) Sensitivity analysis of grid number

(**b**) Sensitivity analysis of time step

**Figure 8.** Sensitivity analysis of grids and time steps.

The numerical uncertainty of resistance coefficient, sinkage, and trim are estimated by the Richardson Extrapolation method. The results are provided in Table 3. As is shown in Table 3, the numerical uncertainty of the resistance coefficient, sinkage, and trim are 3.24%, 4.79% and 4.12%, respectively. The numerical uncertainty is small enough to perform the simulations for the catamaran.

**Table 3.** Calculation of the numerical uncertainty.

|  | $r_k$ | $U_G\%D$ | $U_T\%D$ | $U_{SN}\%D$ |
|---|---|---|---|---|
| $C_t$ |  | 2.14 | 2.41 | 3.24 |
| Sinkage | $\sqrt{2}$ | 4.45 | 1.77 | 4.79 |
| Trim |  | 3.68 | 1.85 | 4.12 |

In Table 3, $r_k$ means the parameter encryption ratio, $U_G$ is the grid uncertainty, and $U_T$ is the time step uncertainty. The numerical uncertainty is

$$U_{SN} = \sqrt{U_G{}^2 + U_T{}^2} \tag{6}$$

Since the uncertainties are small enough, the following simulations are conducted based on the medium grid and the medium time step. The time step is taken as 0.006 s and the grids number of the demihull is 3.14 million. The grid for the bare hull consists of background and demihull. The node distribution and the grids number are shown in Table 4.

**Table 4.** Node distribution and grids number of the bare hull.

| Parts | Number of Nodes in Three Directions | Grids Number (Million) |
|---|---|---|
| Background | $80 \times 46 \times 213$ | 0.784 |
| Demihull | $59 \times 189 \times 66$ | 0.736 |
|  | $137 \times 179 \times 66$ | 1.619 |
| Total |  | 3.140 |

For the self-propulsion catamaran, the grid of the background and the hull are the same as them in the simulations of bare hull. Table 5 presents the node distribution and grids number of each part. The total grids number is about 8.56 million for half of the catamaran.

**Table 5.** Node distribution and grids number of the self-propulsion hull.

| Part | Number of Nodes in Three Directions | Grids Number (Million) |
|---|---|---|
| Background | $80 \times 46 \times 213$ | 0.784 |
| Demihull | $59 \times 189 \times 66$ | 0.736 |
|  | $137 \times 179 \times 66$ | 1.619 |
| Nozzle | $45 \times 45 \times 41$ | 0.083 |
|  | $177 \times 80 \times 41$ | 0.581 |
|  | $177 \times 13 \times 41$ | 0.094 |
| Duct | $128 \times 61 \times 61$ | 0.476 |
|  | $128 \times 241 \times 28$ | 0.864 |
|  | $46 \times 46 \times 32$ | 0.068 |
|  | $46 \times 46 \times 32$ | 0.068 |
| Shaft | $111 \times 181 \times 32$ | 0.643 |
|  | $70 \times 151 \times 33$ | 0.349 |
|  | $39 \times 38 \times 33$ | 0.049 |
|  | $39 \times 38 \times 33$ | 0.049 |
| Stator | $8 \times (119 \times 71 \times 31)$ | $8 \times 0.262$ |
| Total |  | 8.560 |

### 4.2. Numerical Results of Bare Hull

The comparisons between the numerical results and the experimental results of the bare hull are shown in Figures 9–11. According to the experimental data, the resistance coefficient and trim from the two institutes are almost the same and the CFD results have great agreement with them. As to the sinkage, the experimental data from INSEAN and BSHC have the same trend but the values vary from each other obviously, especially at high speed. It is satisfied that the CFD results fall near the data from the two institutes.

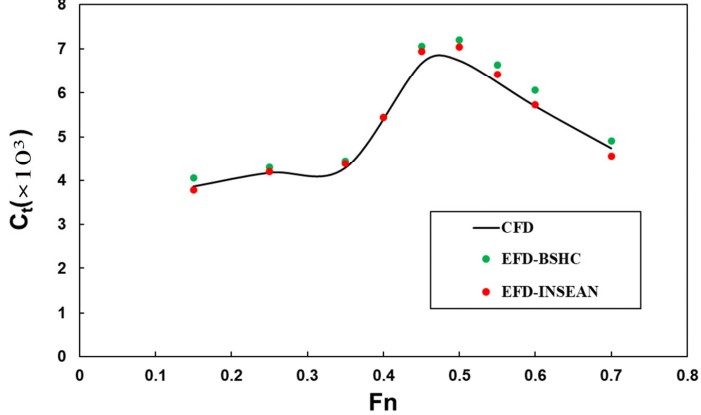

**Figure 9.** Comparison of the resistance coefficient between CFD and EFD results.

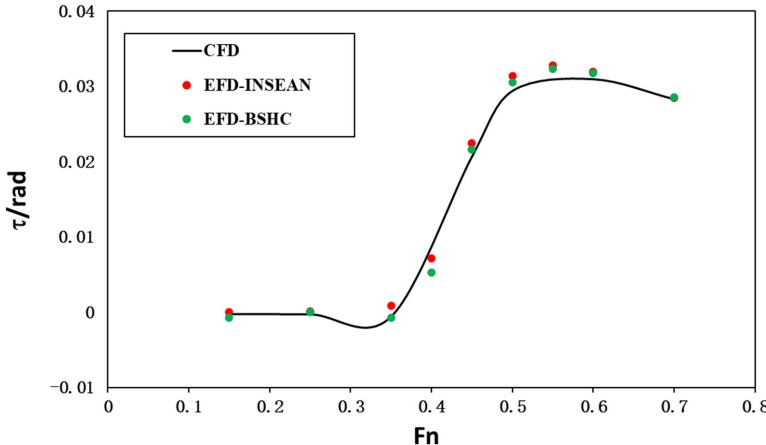

**Figure 10.** Comparison of the trim between CFD and EFD results.

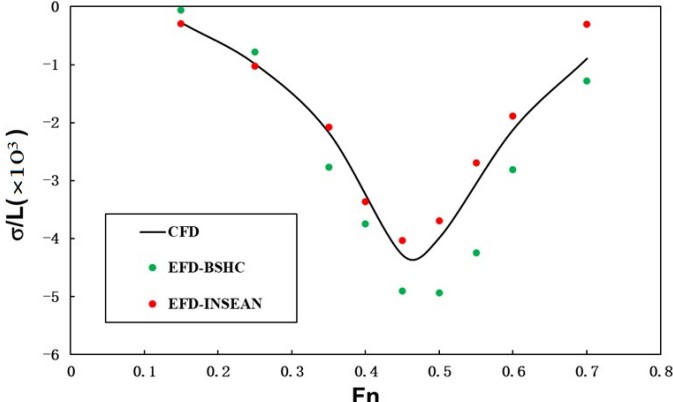

**Figure 11.** Comparison of the sinkage between CFD and EFD results.

### 4.3. Numerical Results of Self-Propulsion Hull

Figures 12 and 13 show the sinkage (measured at the center of gravity) and trim at different Froude numbers of the self-propulsion catamaran. The simulation results are compared with the experimental results from INSEAN and BSHC. The experimental results from the two institutions are different. The CFD results are close to one of the experimental data or between the value from the two institutes. Moreover, the trends of the simulation results agree with them. With the increasing Froude number, the trim increases first and then decreases. The maximum value of trim appears between Fn = 0.55 and Fn = 0.6, and the maximum value is about 2.6°. The sinkage also increases first and then decreases. The maximum value of sinkage appears at Fn = 0.45, and the value is 0.0055 $L_{pp}$.

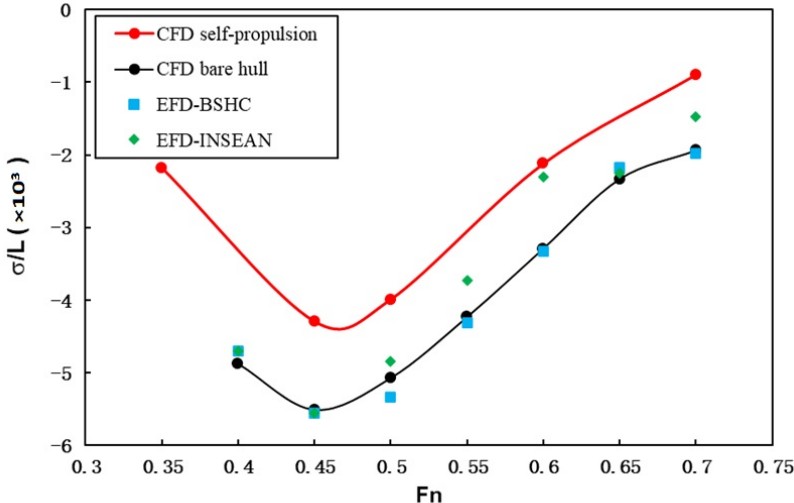

**Figure 12.** The comparison of sinkage between CFD and EFD results for self-propulsion catamaran.

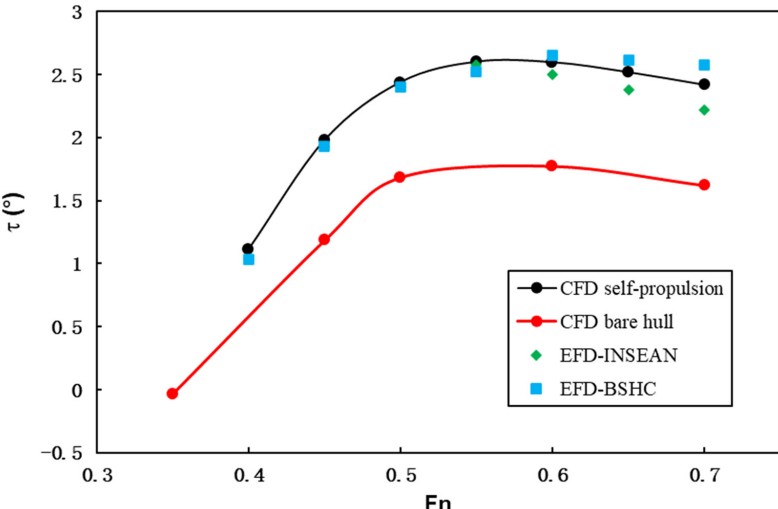

**Figure 13.** The comparison of trim between CFD and EFD results for self-propulsion catamaran.

## 5. Discussion about the Results

### 5.1. Analysis of the Wave around the Hull

Figure 14 shows the wave pattern around the hull. The comparisons of free surface wave height between the bare hull the self-propulsion hull is shown. The wave height on the center plane of the catamaran (Y = 0) and the wave height on the center plane of the demihull (Y = 0.11675 $L_{PP}$) are provided quantitatively. The initial x-position of the bow is 0 and the initial x-position of the stern is 1. Figure 15 shows the streamlines in the inner of the waterjet propulsor.

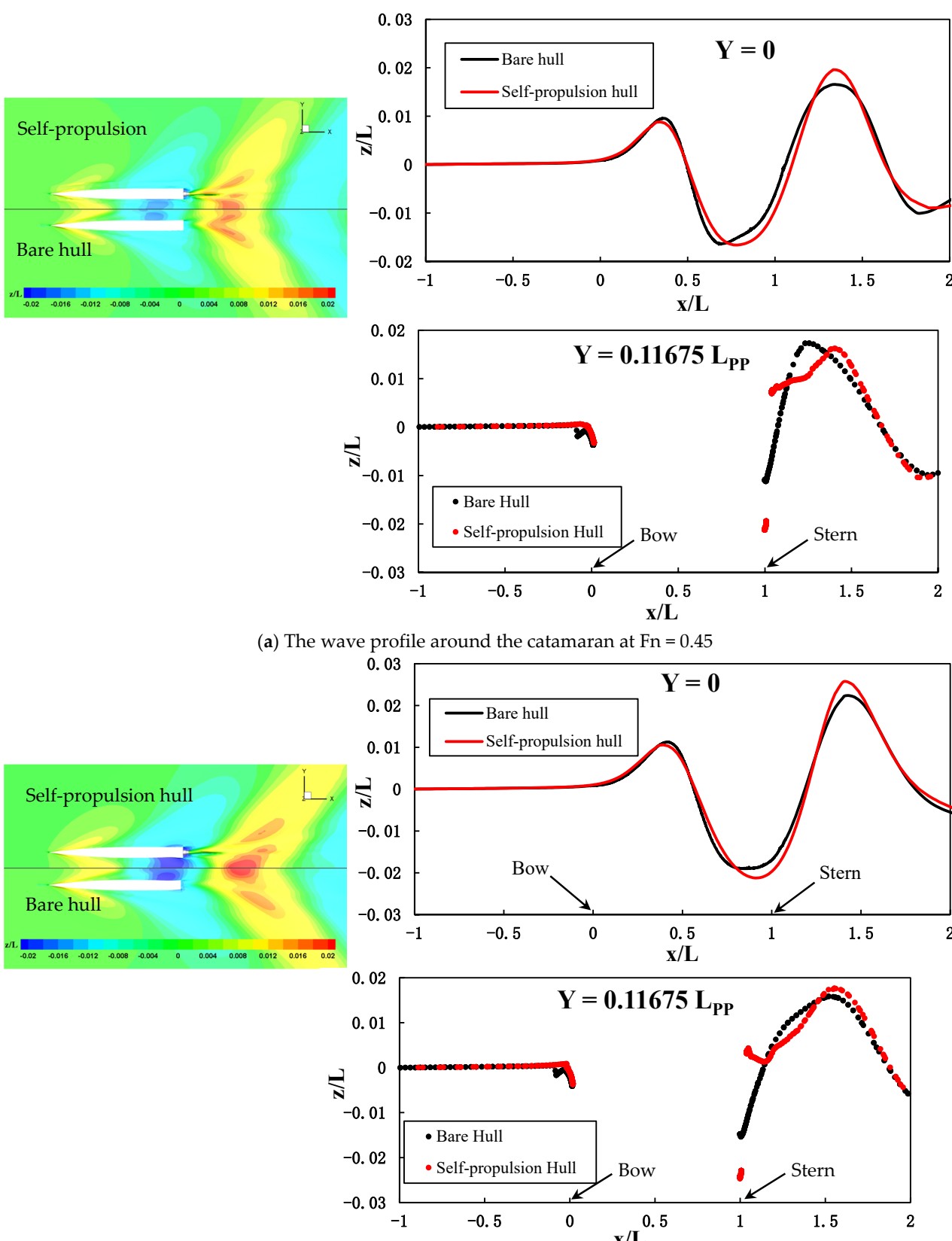

(**a**) The wave profile around the catamaran at Fn = 0.45

(**b**) The wave profile around the catamaran at Fn = 0.5

**Figure 14.** *Cont*.

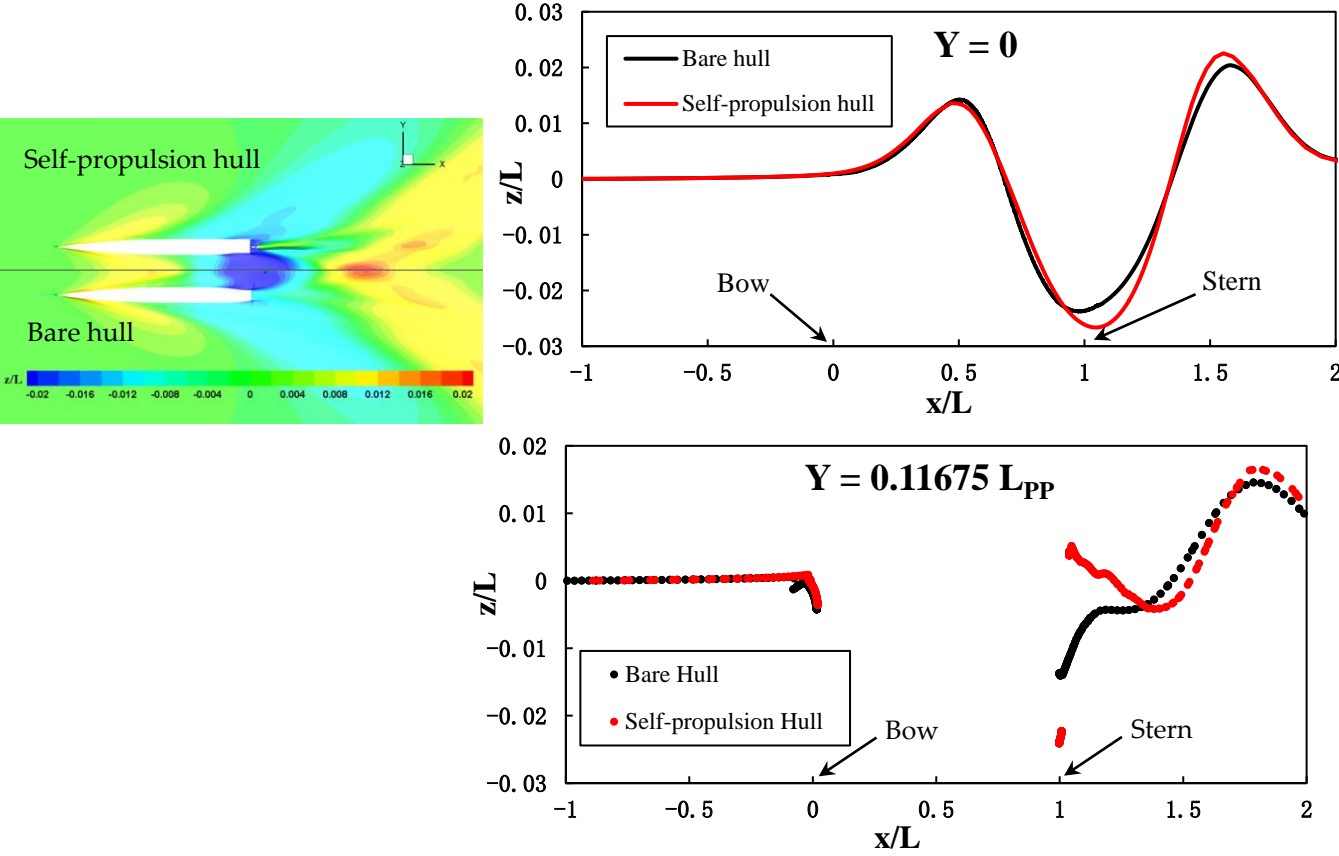

(**c**) The wave profile around the catamaran at Fn = 0.6

**Figure 14.** Wave profile comparison between bare hull and self-propulsion hull.

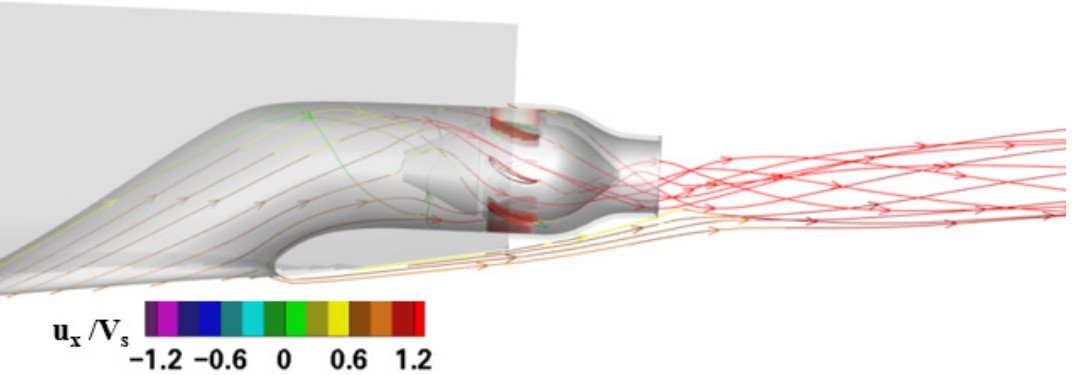

**Figure 15.** The streamlines in the inner of the waterjet propulsor.

According to Figure 14, there is a peak in the front half of the hull, both the bare hull and the self-propulsion hull, on the central plane of the catamaran. In the rear half of the catamaran, there troughs. With the increasing in the Froude number, the peak and trough move back gradually.

Due to the suck effect of the waterjet, the water surface near the stern of the self-propulsion hull is lower than the bare hull. It can be observed from the contours of Figure 14 that the dark area between the demi-body is larger in the top half of the picture. For the center plane of the catamaran (Y = 0), the water surface near and just behind the stern is lower in the self-propulsion condition. With the increasing in the Froude number, the suck effect is more obvious and the water surface is lower and lower in the self-propulsion condition. For the center plane of the demihull (Y = 0.11675 $L_{PP}$), the water

surface behind the transom board is significantly lower in the self-propulsion condition because of the suck effect. With the influence of flow from the nozzle, the water surface rises and changes the wave behind it. However, the region influenced by the flow from the nozzle is away from the hull, and the impact on the hull is minor.

### 5.2. Analysis of the Hull Attitude

With the operation of the waterjet system, the attitude of hull is affected obviously. Figures 12 and 13 compare the trim and sinkage of the bare hull and self-propulsion hull, respectively. With the increasing in the Froude number, the changes in hull attitude are consistent in the two cases, which means that the waterjet does not have a decisive impact on the hull motion.

According to Figures 12 and 13, the absolute values of trim and sinkage are both greater in self-propulsion conditions, which are associated with the pressure distribution on the bottom of hull. The pressure on the bottom of the hull is extracted and the position of the extracted pressure is shown in Figure 16. Figure 17 shows the pressure distribution on the hull bottom at Fn = 0.45 and 0.6. Because of the suck effect of the waterjet, the flow is accelerated from the position in front of the catamaran. In the meantime, the pressure decreases and varies from the bare hull.

According to Figure 17, the pressure distribution varies a lot especially on the bottom of the stern near the duct. Thus, the trim by the stern exacerbates. With the decrease in pressure, the self-propulsion hull sinks, which results in a more obvious sinkage than the bare hull.

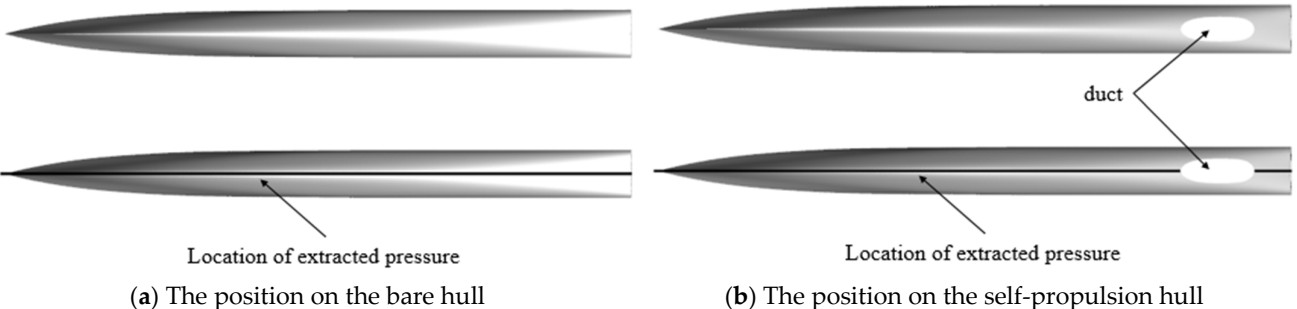

(**a**) The position on the bare hull          (**b**) The position on the self-propulsion hull

**Figure 16.** The position of the extracted pressure.

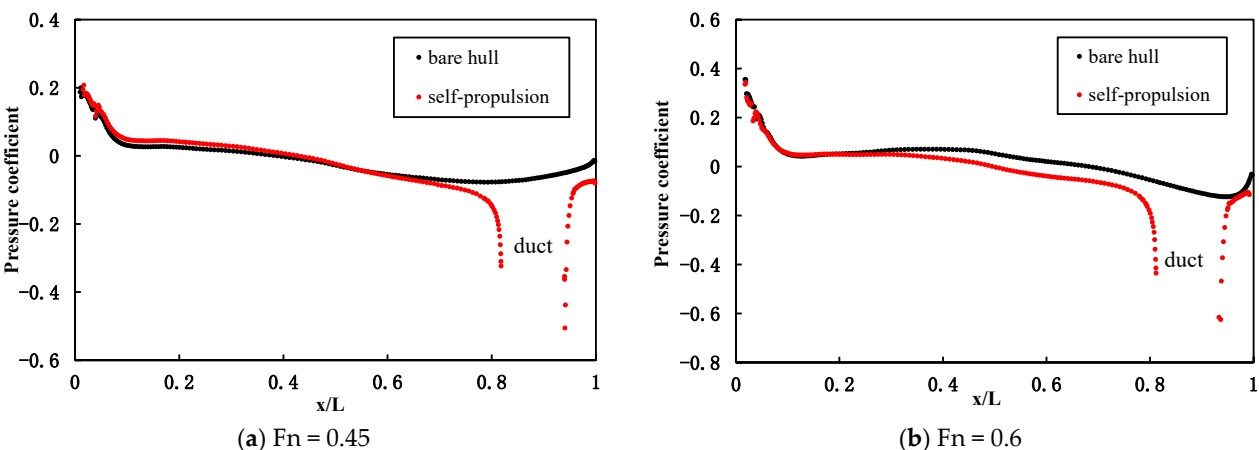

(**a**) Fn = 0.45                              (**b**) Fn = 0.6

**Figure 17.** Comparison of the pressure coefficient on the center plane of hull bottom.

The pressure distribution is associated with the velocity near the hull. Figure 18 presents the velocity near the hull at Fn = 0.6. According to the velocity field, it is easy to find that the duct influences the flow velocity near the stern of the hull. With the development of the flow around the hull bottom, the boundary layer of the bare hull

becomes thicker and thicker. The self-propulsion hull is the same in front of the intake tangency point. However, the duct interrupts the development of the boundary layer, which leads to the redevelopment of the boundary layer behind the duct. Figure 19 presents the z-position of the hull bottom and the boundary layer near the bottom at different x-positions. The boundary layer is defined as the position where the velocity is 0.9 times of the incoming flow. The black curves are for the bare hull while the red curves are for the self-propulsion hull. The hull attitudes in the two conditions are different, so the position of the hull is different. There is an intake of the duct at x/L ranging from 0.81 to 0.93, so the surface of the bottom and boundary layer is not provided. At any x-position, the region of z-position between two curves represents the thickness of the boundary layer. For the self-propulsion hull, the thickness of the boundary layer changes obviously near the waterjet, which is the main difference from the bare hull. The boundary layer near the duct is discontinuous with the position ahead and redevelops in the rear of the duct. It leads to the thickness of the boundary layer changing obviously, which changes the flow velocity and the pressure. It has a great impact on the attitude of the hull.

In addition, the change and difference of the attitude can also be analyzed from the wave around the hull. According to Figure 14, there is a trough near the stern of the catamaran on the center plane of the catamaran. When the stern of the hull is located in the trough, the trim by stern exacerbates. When Fn = 0.6, the x-position of the trough is going to be out of the catamaran, so that its effect on the trim reduces. This is the reason the trim decreases when Fn > 0.6. As for the sinkage, it is also associated with the trough of wave. The absolute value reaches the maximum when the trough is near the center of gravity (when Fr = 0.45). With the increasing speed, the trough of the wave moves back, which has an impact on the sinkage. In addition, the change in the trim also lifts the center of gravity. As for the difference in the two conditions, the wave height near the stern is obviously different from the two conditions, while the wave height differs little when it is away from the waterjet. This leads to the trim and sinkage of the self-propulsion hull being larger than the bare hull.

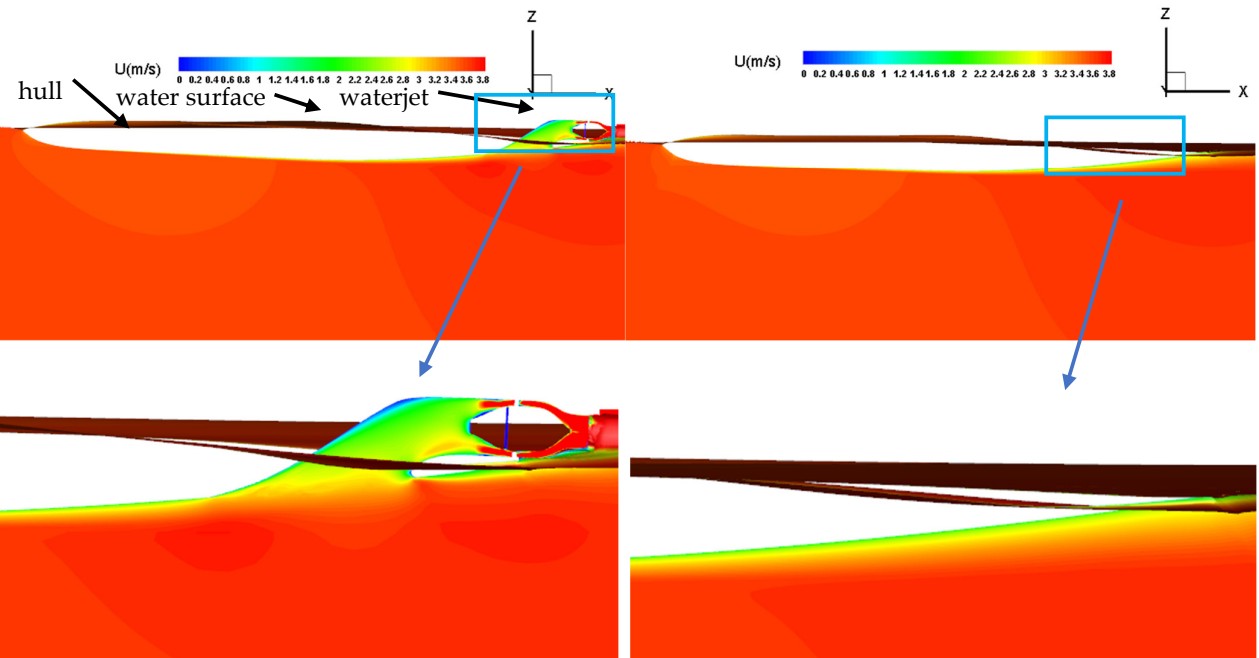

**Figure 18.** The velocity around the stern of the catamaran.

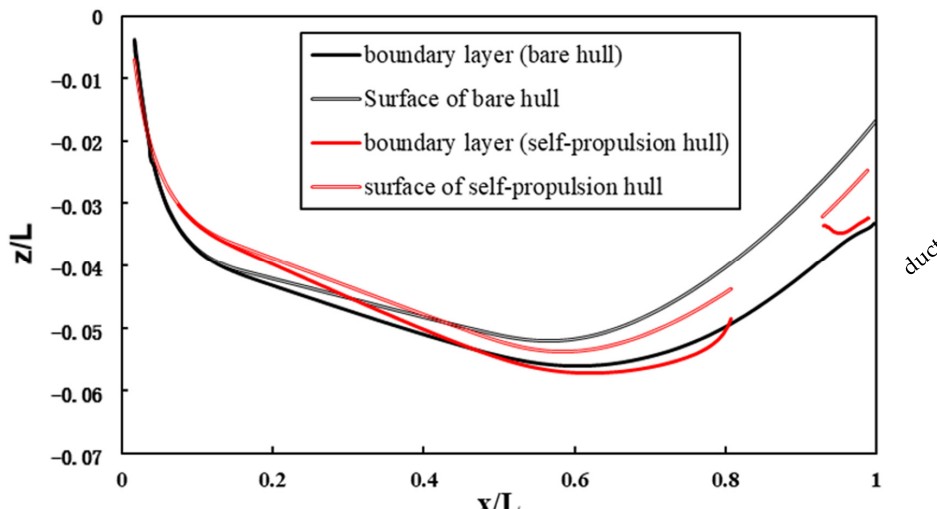

**Figure 19.** The position of the hull surface and boundary layer.

*5.3. Analysis of the Self-Propulsion Efficiency*

To understand the energy conversion efficiency in the self-propulsion waterjet-propelled catamaran, the efficiency is divided into multiple components. The decomposition of the energy conversion is shown in Figure 20.

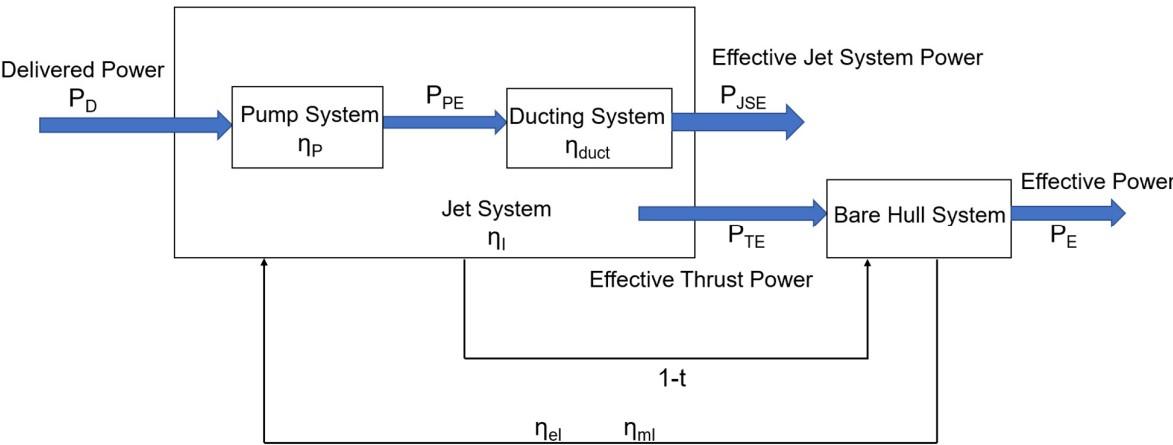

**Figure 20.** Energy conversion between a waterjet system and a hull.

The different stations of the waterjet are presented in Figure 21. It is based on the recommended procedures and guidelines from ITTC [30,31]: "0" is the far field section, "1" is the capture area, "2" is the inlet throat of the flow channel, "3" is the section in front of the rotor, "4" is the section at the rotor, "5" is the section behind the rotor, "6" is the nozzle section, and "7" is the contraction section after the water jet.

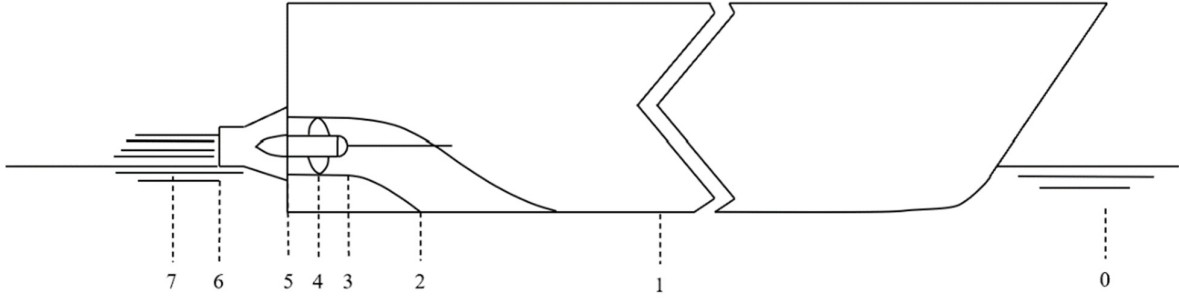

**Figure 21.** A sketch of the different stations of the catamaran.

In order to facilitate the subsequent analysis process, the location and shape of the capture area should be determined first. According to the previous experiments [32], both thrust and power estimations were insensitive to the capture area shape. Only a 1% change in estimated power was produced by a 20% variation in capture area width. At this point, it is anticipated that the derived momentum and energy fluxes in the region of the capture area are insensitive to the minor variations in area shape. As a result, it is feasible to make some assumptions to simplify the shape. The capture area here is simplified to a half elliptical shape in Figure 22 which is introduced by ITTC [32]. The width of the capture area is considered to be 1.5 times the intake geometry width, and the height of the capture area is adjusted according to the flow rate. Based on these, the shape of the capture area is determined initially. The final shape of the capture area is also based on the shape of the hull. According to the ITTC Waterjet Specialist Committee [32,33], the capture area should be placed one inlet width forward of the intake tangency point. The shape of the capture area and the velocity distribution on the capture area at different Froude numbers are shown in Figure 23.

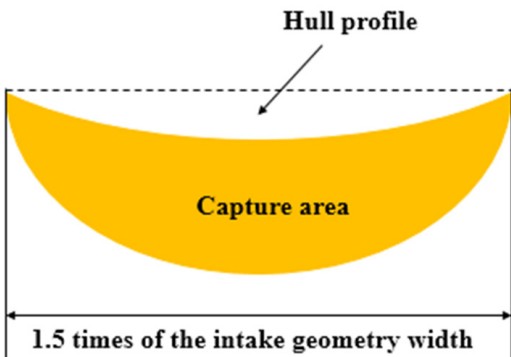

**Figure 22.** The sketch of the capture area.

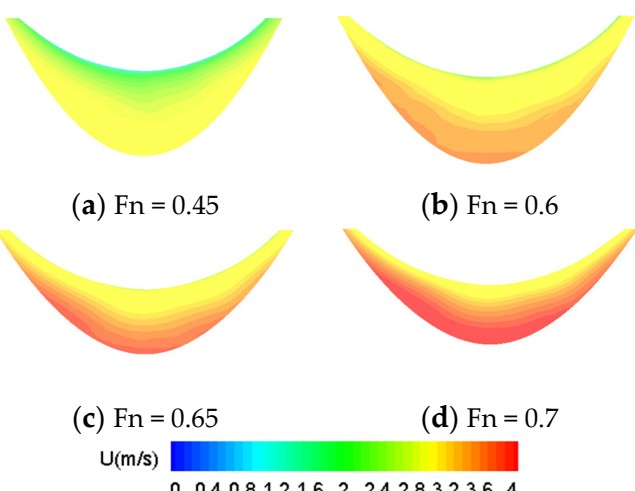

**Figure 23.** The velocity distribution on the capture area.

The volume flow rate in the duct agrees well with the experimental data according to Figure 24. The difference between the simulated volume flow rate and the experimental results provided by the two institutes is less than 4%.

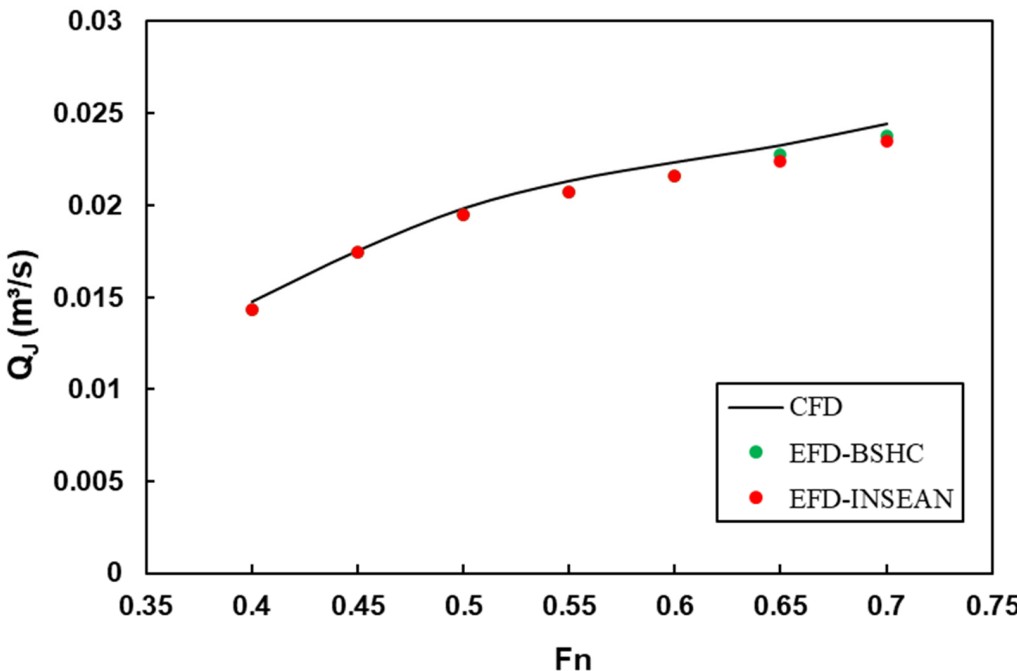

**Figure 24.** The comparison of volume flow between CFD results and EFD data for self-propulsion catamaran.

1. Thrust deduction

The gross thrust $T_{gross}$ is defined as the force vector pertinent to the change in momentum flux ($\triangle \overline{M}_x$) over the selected control volume. According to the ITTC recommended procedures, the thrust exerted by the waterjet on the hull has the following relation with the bare hull resistance:

$$\triangle \overline{M}_x = T_{gross} = \frac{R_{BH}}{1 - t} \tag{7}$$

The change in momentum flux can be determined from station 1 and station 6:

$$\triangle \overline{M}_x = \overline{M}_{x6} - \overline{M}_{x1} \tag{8}$$

For any section,

$$\overline{M}_x = \iint \rho u_x^2 dA \tag{9}$$

Figure 25 compares the gross thrust and bare hull resistance at different Froude numbers. It also compares the experimental data with the simulation results. The thrust deduction provided in Figure 26 is calculated from the bare hull resistance and gross thrust. The differences of gross thrust between the CFD results and EFD results are less than 4%. The thrust deduction ranges from 0.1 to 0.2, which means the waterjet system has a negative effect on the hull.

This is associated with the wave near the transom board. In the self-propulsion condition, the water surface is lower than the bare hull because of the suction effect and the location of the nozzle, which makes the self-propulsion hull submerged in the water less than the bare hull. The pressure on the transom board tends to make the resistance reduce. Thus, it tends to a positive thrust deduction. Figure 27 shows the immersed transom area. The water surface is rendered in color. In addition, the self-propulsion hull tends to a greater trim and sinkage, which means a larger wetted surface. It also leads to a positive thrust deduction.

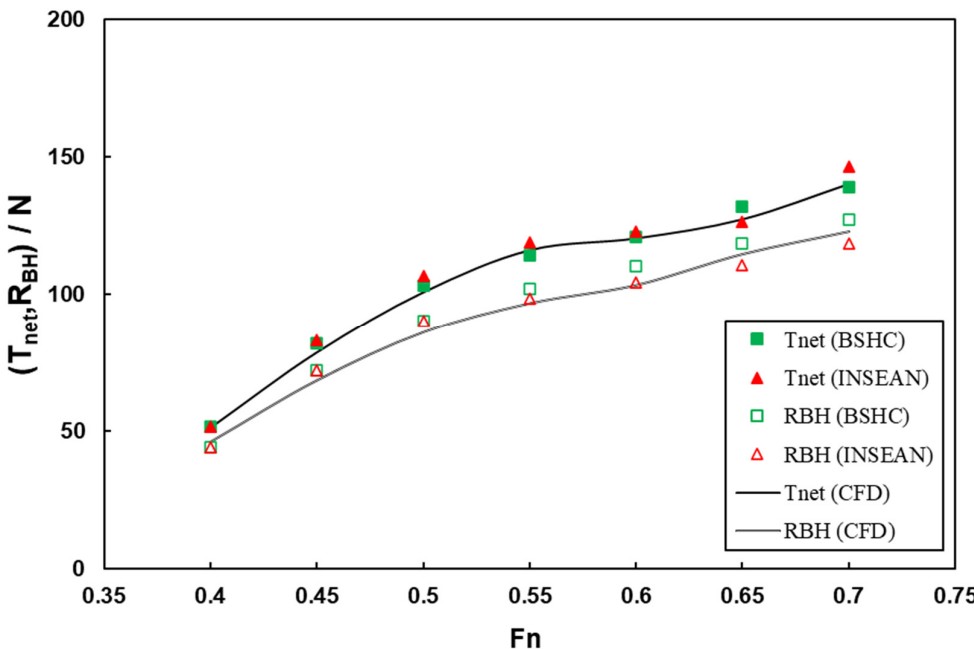

**Figure 25.** Comparison of the gross thrust and bare hull resistance between EFD and CFD.

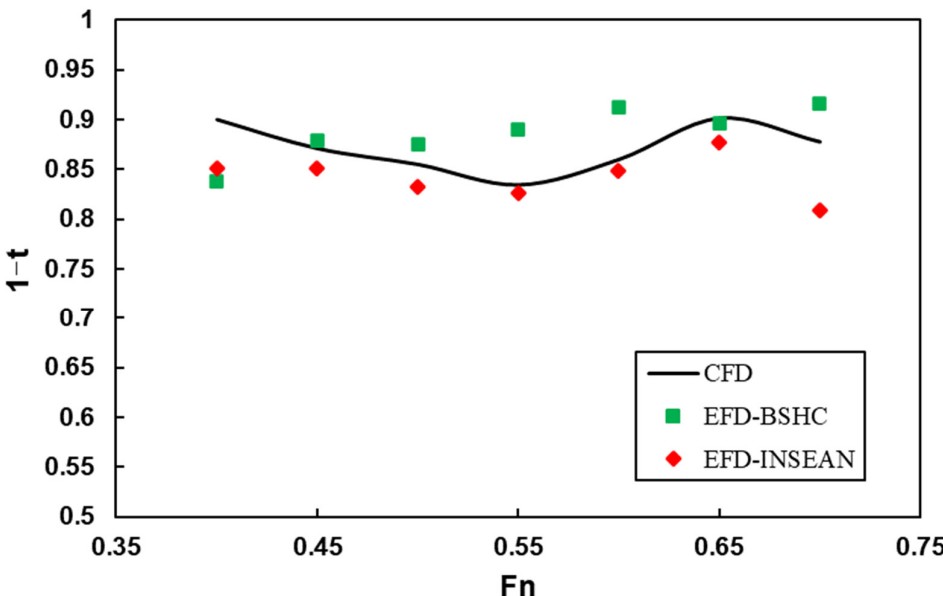

**Figure 26.** Comparison of the thrust deduction between EFD and CFD.

2.  The free stream efficiency

The free stream efficiency $\eta_0$ is defined as

$$\eta_0 = \eta_I \eta_{duct} \eta_p \tag{10}$$

$\eta_I$ is the ideal efficiency which is defined as

$$\eta_I = \frac{2}{NVR + 1} \tag{11}$$

where $NVR = \frac{V_j}{V_s}$. $V_j$ means the flow speed at the nozzle and $V_s$ means the ship speed.

Pump efficiency:

$$\eta_p = \frac{P_{PE}}{P_D} \tag{12}$$

Ducting efficiency:

$$\eta_{duct} = \frac{P_{JSE}}{P_{PE}} \tag{13}$$

In the formulas, the effective pump power:

$$P_{PE} = \rho g Q_J H_{35} \tag{14}$$

The effective jet system power:

$$P_{JSE} = E_6 - E_1 \tag{15}$$

$H_{35}$ means the pump head.

$P_D$ is the power received by the rotor:

$$P_D = 2\pi n Q \tag{16}$$

For any station $j$, the axial energy flux through a cross-sectional area $A_s$ at station s is defined as:

$$E_s = \int \rho \left( \frac{1}{2} u_i^2 + \frac{p}{\rho} + g z_s \right) (u_i n_i) dA \tag{17}$$

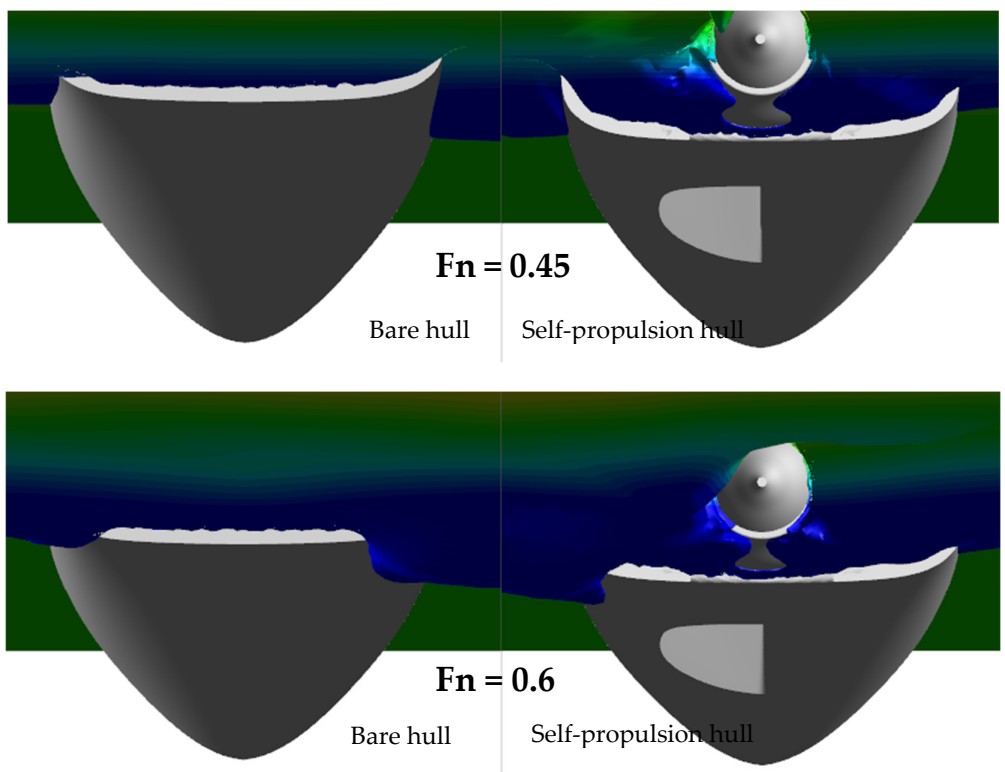

**Figure 27.** Comparison of the immersed transom area of hull.

Figure 28 presents the distribution of non-dimension axial velocity on the nozzle section (station 6). With the increase in Froude number, the nozzle velocity ratio (NVR) first increases and then decreases. Thus, the ideal efficiency first decreases and then increases.

Figure 29 presents the duct efficiency, ideal efficiency, and pump efficiency. Then, multiplying them obtains the free stream efficiency. The duct efficiency is always higher than 0.9, which means the loss caused by the duct accounts for a small proportion. The pump efficiency ranges from 0.45 to 0.5, which is obviously lower than the other two components of the free stream efficiency.

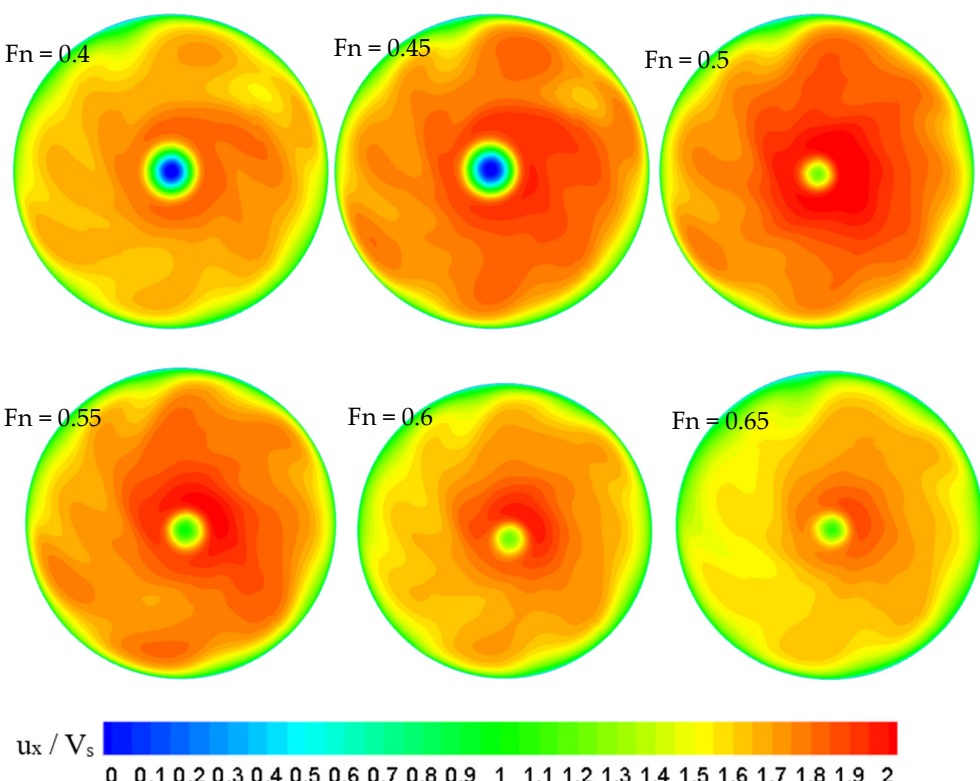

**Figure 28.** The axial velocity ($u_x/V_s$) distribution on the nozzle section.

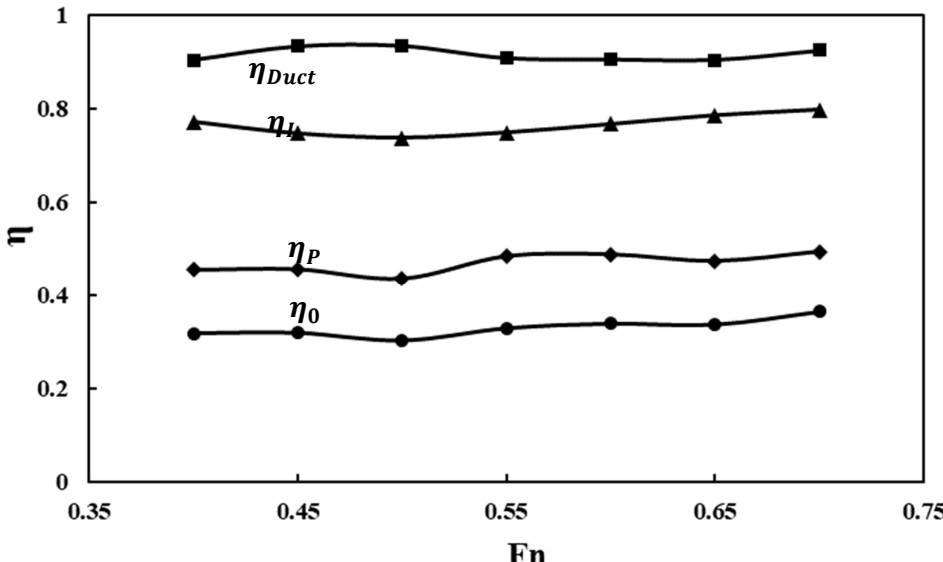

**Figure 29.** The results of free stream efficiency.

3.  The overall efficiency

The overall efficiency can be divided into free stream efficiency and interaction efficiency as

$$\eta_D = \eta_0 \eta_{INT} \tag{18}$$

$\eta_{INT}$ is the interaction efficiency:

$$\eta_{INT} = (1 - t)\frac{\eta_{eI}}{\eta_{mI}} \tag{19}$$

$\eta_{mI}$ is the momentum interaction efficiency; and $\eta_{eI}$ is the energy interaction efficiency. For $\eta_{mI}$, it satisfies the following relationship:

$$\frac{1}{\eta_{mI}} = \frac{P_{TE}}{P_{TE0}} = 1 + \frac{1 - IVR}{NVR - 1} \tag{20}$$

where $NVR = \frac{V_j}{V_s}$ and $IVR = \frac{V_{in}}{V_s}$. $V_j$ means the flow speed at the nozzle, $V_{in}$ means the flow speed at the capture area, and $V_s$ means the ship speed. $P_{TE}$ means the effective thrust power and $P_{TE0}$ is the effective thrust power in the free stream condition.

For $\eta_{eI}$, it has the relationship with effective jet system power and effective jet system power in the free stream condition as the following formula:

$$\eta_{eI} = \frac{P_{JSE0}}{P_{JSE}} = \frac{E_6 - E_0}{E_6 - E_1} \tag{21}$$

Figure 30 presents the results of three different efficiency. The interaction efficiency is always around 0.8, which means there is a negative effect to the waterjet due to the waterjet–hull interaction. Thus, the overall efficiency of the waterjet system is always smaller than the free stream efficiency. According to the results, the overall efficiency of the waterjet system is mainly determined by the free stream efficiency. The numerical overall efficiency agrees well with the experimental data according to Figure 31. The difference of the overall efficiency between the CFD results and the average EFD results from the two institutions is less than 7%.

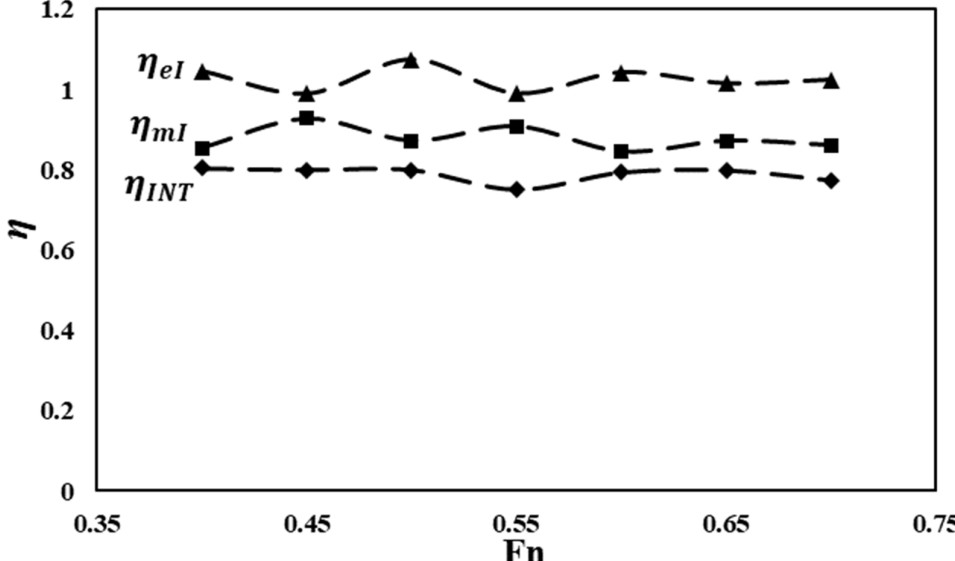

**Figure 30.** The results of the interaction efficiency and overall efficiency.

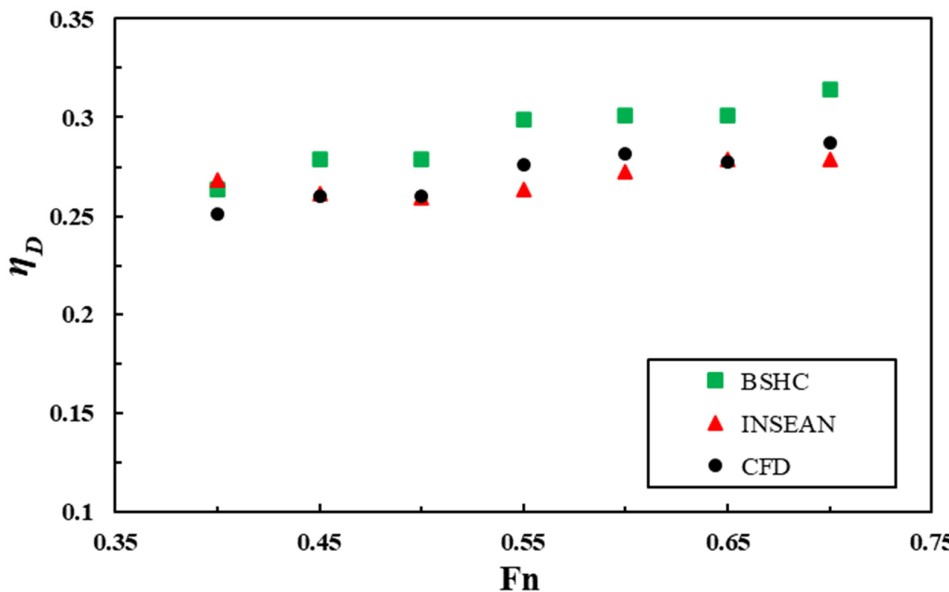

**Figure 31.** Comparison of the overall efficiency between EFD and CFD.

## 6. Conclusions

This paper presents the research about a waterjet propelled catamaran. The simulations of the bare hull and self-propulsion hull are conducted using an in-house solver. The pump effect of the rotor is simulated by a body force model to save computing costs of the simulations. Moreover, the simulation results of the bare hull and self-propulsion hull are compared in detail to analyze the hull effect on the waterjet system. In addition, the characteristic of the waterjet system as well as the waterjet–hull interaction are analyzed according to the ITTC recommended procedures. The conclusions are as follows:

1. The numerical uncertainty of the resistance coefficient, sinkage, and trim are 3.24%, 4.79%, and 4.12%, respectively. The numerical uncertainty is small enough to perform further simulations for the catamaran. Moreover, the validations are performed by comparing the CFD results with the experimental data from INSEAN and BSHC. The good match represents the accuracy of the CFD solver and numerical method.

2. There are two reasons for the greater trim and greater sinkage when the waterjet operated behind the catamaran: On the one hand, the water surface near the stern is lower with the operation of the waterjet, while the water surface near the bow does not change much. On the other hand, the flow near the waterjet is accelerated by the suction effect of the pump, which reduces the pressure distribution on the bottom of the hull near the stern. The two aspects have a great impact on the attitude of the hull.

3. Because of the larger wetted surface and the smaller wetted transom board, the thrust deduction is positive within the scope of the study, which means the waterjet system has a negative effect on the hull. Moreover, the interaction is also negative for the efficiency of the waterjet. As a result, the overall efficiency of the waterjet system behind the hull is about 0.75~0.8 times the free stream efficiency.

4. Among the components of the overall efficiency (including ducting efficiency, ideal efficiency, pump efficiency, and interaction efficiency), the ducting efficiency is the highest and the pump efficiency is the lowest. Moreover, the ducting efficiency is higher than 0.9, which means the loss on the duct is relatively small. However, the pump efficiency is significantly lower than the other components. In addition, the interaction between the hull and waterjet system is also an important part, and a positive interaction is a desired goal.

**Author Contributions:** Methodology, D.F.; investigation, Y.Z. and W.D.; resources, D.F. and H.Z.; data curation, Y.Z. and W.D.; writing—original draft, Y.Z.; writing—review and editing, Y.Z. and J.Y.; visualization, Y.Z.; supervision, D.F., W.D., J.Y. and H.Z.; project administration, D.F.; funding acquisition, D.F. All authors have read and agreed to the published version of the manuscript.

**Funding:** This work was supported by the China National Science Foundation YEQISUN Joint Funds under grant U2141228.

**Institutional Review Board Statement:** Not applicable.

**Informed Consent Statement:** Not applicable.

**Data Availability Statement:** Not applicable.

**Conflicts of Interest:** The authors declare no conflict of interest.

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
