# Peer review of "Numerical Study on the Waterjet–Hull Interaction of a Free-Running Catamaran"

_jmse, doi:10.3390/jmse11040864_

Round 1

Reviewer 1 Report

This paper uses CFD in-house code to simulate the interaction between a waterjet system and a catamaran.. 

In the abstract authors mention RANS, while in many places I see that the time analysis was conducted hence URANS.

The authors need to explicitly mention the novelty of the study. Is this the first kind of validation against experimental results for such analysis? What is novel about this research? Does this work contribute on top of the experimental work shown in this paper? If so, please explicity describe this in the ABSTRACT,

The CFD simulations are done generally very well, following sensible and practical guidelines. However, it would be nice to see grid convergence plots, residual plots, l2 norm plots etc to raise confidence in this simulation which will improve this paper as I can see that the validation against experiment is quite rigid.

Authors also need to re-read the paper and improve english, especially in the conclusion. Furthermore, the legends given are very short, a more scientific and detailed legend should be provided, wherever possible.

L235 - "E means the test data from INSEAN and S means the simulation results" should be written more formally,  For instance, " E - test data from INSEAN and S - simulation results"

* It would be better to have the time step and grid uncertainty as a plot rather than a table. The jump between the 3 levels in terms of percentage is still quite high and I was not convinced whether a full grid or time step independence has been achieved but then Fig 8-10 shows the CFD results are in close agreement with exp for a wide Fn. 

* Fig 13 a,b and c what are the contours on the left hand side representing? the legend is not clear at all. The legend should be scaled up and possibly put to the side or below the contour.

Reviewer 2 Report

Overall, the paper is written well. However, one point regarding the using symmetry boundary condition to model turbulent interaction of two jets seems inappropriate. I understand the authors have shown comparison with experimental data and it confirms their method agrees well. However, it is not physically appropriate. If the authors could provide one case where they show keeping the full domain doesn't change the results especially for the plots shown in Figure 13 that would be a good.

Reviewer 3 Report

Shape recommendations

1. Table 1, your data does not maintain the same number of significant figures. Too many decimal places no longer have significant value. It is recommended to present all with the same number of significant figures.

2. Figure No. 2 must contain indications about the parts shown for a better understanding of the readers.

3. Figure 4 I recommend that the vertical scale go from 0.3 to 0.9 to avoid areas without results that do not contribute to the presentation of results.

4. Figures No. 5 and No. 6 must contain information that better explains the sequence of images.

5. Table 3 and 4 present the results with the same numbers of significant figures. For example the Trim of 1.8 should be 1.800. Also review all the tables.

6. Figure No. 8 I recommend that the vertical scale go from 0.003 to avoid areas without results that do not contribute to the presentation of results. Additionally, a lot of decimals do not look good and it is better to put 3,4,5,6... on the scale and indicate in the vertical that it is (10^-3.) Likewise, review other graphs.

Content recommendations

7. On line 98 of the introduction, detail more why you consider that the proposed method has results with uncertainty.

8. At the end of the introduction, include a flowchart of how the manuscript has been distributed for a better understanding of the reader.

9. Equation (4) needs to be better explained in more detail.

10. Lack of detail on the values of the Grids applied in the discretization, sizes, shapes, among others.

11. The conclusions should include a comment on the uncertainty of the results.

12. There are very old references, change them for younger ones. (recommended no more than 20 -15 years)

Reviewer 4 Report

The authors presented a Numerical study on the waterjet-hull interaction of a free-running catamaran.

The main quantitative findings are to be mentioned in the abstract.

The novelty of the paper is to be clearly stated.

The authors used an in-house numerical code; thus, the numerical method is to be described in details.

What is the used numerical method ?

How is the mesh generated?

How is the discretization performed?

What is the convergence criterion?

What are the characteristics of the used computer?

What is the time needed for the convergence?

In addition to the presented quantitative validation, a qualitative validation/verification (flow structure for example) is to be performed.

The dimensions of the computational domain are to be justified.

The used turbulence model is to be justified.

Fig 14, has low resolution.

The paper is to be checked against misprints and grammatical mistakes.

Round 2

Reviewer 4 Report

After revision, the paper can be accepted for publication